# Activation of G protein-coupled estrogen receptor signaling inhibits melanoma and improves response to immune checkpoint blockade

Christopher A Natale[1], Jinyang Li[2], Junqian Zhang[1], Ankit Dahal[1], Tzvete Dentchev[1], Ben Z Stanger[2], Todd W Ridky[1]*

[1]Perelman School of Medicine, Department of Dermatology, University of Pennsylvania, Philadelphia, United States; [2]Abramson Family Cancer Research Institute, Perelman School of Medicine, University of Pennsylvania, Philadelphia, United States

**Abstract** Female sex and history of prior pregnancies are associated with favorable melanoma outcomes. Here, we show that much of the melanoma protective effect likely results from estrogen signaling through the G protein-coupled estrogen receptor (GPER) on melanocytes. Selective GPER activation in primary melanocytes and melanoma cells induced long-term changes that maintained a more differentiated cell state as defined by increased expression of well-established melanocyte differentiation antigens, increased pigment production, decreased proliferative capacity, and decreased expression of the oncodriver and stem cell marker c-Myc. GPER signaling also rendered melanoma cells more vulnerable to immunotherapy. Systemically delivered GPER agonist was well tolerated, and cooperated with immune checkpoint blockade in melanoma-bearing mice to dramatically extend survival, with up to half of mice clearing their tumor. Complete responses were associated with immune memory that protected against tumor rechallenge. GPER may be a useful, pharmacologically accessible target for melanoma.
DOI: https://doi.org/10.7554/eLife.31770.001

*For correspondence:
ridky@pennmedicine.upenn.edu

## Introduction

Melanoma is the most deadly form of skin cancer and incidence is rising worldwide. Despite recent advancements in immunotherapies, the majority of patients with metastatic melanoma will still succumb to their disease (*Hamid et al., 2013*; *Ribas et al., 2016*). There is an acute need for new therapeutic strategies that augment the efficacy of standard-of-care immune checkpoint inhibitors. Clues to potential new therapeutic targets for melanoma may be found in 50 year old observations (*White, 1959*), validated in recent studies, that female sex, history of multiple pregnancies, and decreased maternal age at first birth are associated with decreased melanoma incidence and favorable prognosis (*Bannister-Tyrrell et al., 2015*; *Gandini et al., 2011*; *Hersey et al., 1977*; *Joosse et al., 2013*; *Karagas et al., 2006*; *Magnus, 1977*). Although the mechanism of this protective effect is unknown, the clinical association suggests that sex hormone signaling is involved. We hypothesized that understanding the relevant hormones, receptors, and downstream signaling events activated in melanocytes by pregnancy-associated sex steroids would help define the mechanism of the female melanoma protective effects, and suggest new therapeutic opportunities.

In melanocytes, facultative pigmentation and differentiation is primarily regulated by melanocortin receptor 1 (MC1R), which is a $G_s$-coupled G protein-coupled receptor (GPCR). MC1R activation results in the stimulation of adenylate cyclase, which produces cyclic adenosine monophosphate

(cAMP). cAMP activates a range of diverse downstream pathways, including the exchange protein directly activated by cyclic AMP (EPAC) (*Lissitzky et al., 2009*). In melanocytes, cAMP activates protein kinase A (PKA), which phosphorylates and activates the cAMP response element-binding protein (CREB). CREB is a component of a transcription factor complex that drives transcription of microphthalmia-associated transcription factor (MITF), which is known as the master regulator of melanocyte differentiation (*D'Orazio and Fisher, 2011*). MITF directs transcription of melanocyte specific genes required for melanin synthesis including tyrosinase. In previous studies we determined that estrogen, which is higher in females, especially during pregnancy, acts directly on skin melanocytes to increase both pigment production and melanocyte differentiation (*Natale et al., 2016*). These estrogen effects are mediated entirely through a GPCR named G-protein coupled estrogen receptor (GPER). GPER activates signaling pathways that are completely distinct from classical estrogen receptors (*Filardo et al., 2002*). Although there are no approved drugs that specifically target GPER, we determined that GPER is activated in both female and male normal melanocytes by estrogen, as well as by a selective agonist (G-1) that activates GPER signaling without affecting the activity of classical estrogen receptors (ERα/β) (*Bologa et al., 2006*). An independent laboratory subsequently validated these results (*Sun et al., 2017*). Here, we show that GPER activation in melanoma induces a constellation of long-lasting phenotypic changes that inhibit tumor growth, and also render tumor cells more susceptible to clearance by native immune cells, which increases the clinical efficacy immune checkpoint blockade. Selective GPER agonists may represent a new class of anticancer therapeutics.

## Results

To test whether pregnancy affects melanoma development, we used genetically-defined human melanoma (heMel) xenografts (*Chudnovsky et al., 2005*; *McNeal et al., 2015*). In this tissue model, primary human melanocytes were engineered with lentiviruses to express mutant oncoproteins commonly associated with spontaneous human melanoma (*McNeal et al., 2015*) including BRAF$^{V600E}$ (doxycycline-inducible), dominant-negative p53$^{R248W}$, active CDK4$^{R24C}$ and hTERT (*Figure 1—figure supplement 1A*). The oncogene expressing melanocytes were combined with primary human keratinocytes and native human dermis to construct functional 3-dimensional human skin tissues that were grafted into the orthotopic location on the backs of female mice (*Figure 1—figure supplement 1B*). After grafts healed, mice were randomized and separated into nonbreeding or breeding groups (*Figure 1A*). Doxycycline chow was then provided to induce the BRAF$^{V600E}$ oncogene in all animals. After 15 weeks and three consecutive pregnancies in the breeding group (or no pregnancies in the nonbreeding group), human tissues were harvested and analyzed histologically. Grafts from the nonbreeding group developed into melanocytic neoplasms with hallmark features of human melanoma including large, mitotically active melanocytic nests with cellular atypia (*Figure 1B–D* and *Figure 1—figure supplement 1C*). In contrast, tissues from the breeding group were relatively unremarkable, and contained primarily quiescent, single, non-proliferating melanocytes that were confined to the basal epidermal layer. These results show that repeated pregnancies inhibit the growth of BRAF-driven human melanocytic neoplasia.

The primary role of a fully differentiated epidermal melanocyte is to produce melanin pigment that protects the skin from ultraviolent radiation (*D'Orazio and Fisher, 2011*; *D'Orazio et al., 2006*; *Lin and Fisher, 2007*). As with most cell types, melanocyte differentiation and proliferation are inversely correlated, and melanocytes in normal skin rarely proliferate outside of cycling hair follicles (*Jimbow et al., 1975*; *Rabbani et al., 2011*). Melanoma tissue is generally less differentiated than normal melanocytes or benign nevi. In our xenograft studies, pregnancy was associated with the relative lack of proliferating melanocytes and a corresponding increase in epidermal melanin, suggesting that these melanocytes are relatively more differentiated. Although the nonbreeding group, which developed melanomas, had significantly more melanocytes in the grafted skin than the breeding group, melanin abundance within the surrounding epidermal keratinocytes was reduced (*Figure 1E*). These data suggest that pregnancy inhibits melanoma development and induces melanocyte pigment production.

To test whether pregnancy-associated hormones induce long-lasting changes in melanocytes that could account for the melanoma survival benefit observed in some studies of women who experienced pregnancy decades earlier, we transiently exposed primary human melanocytes to estrogen

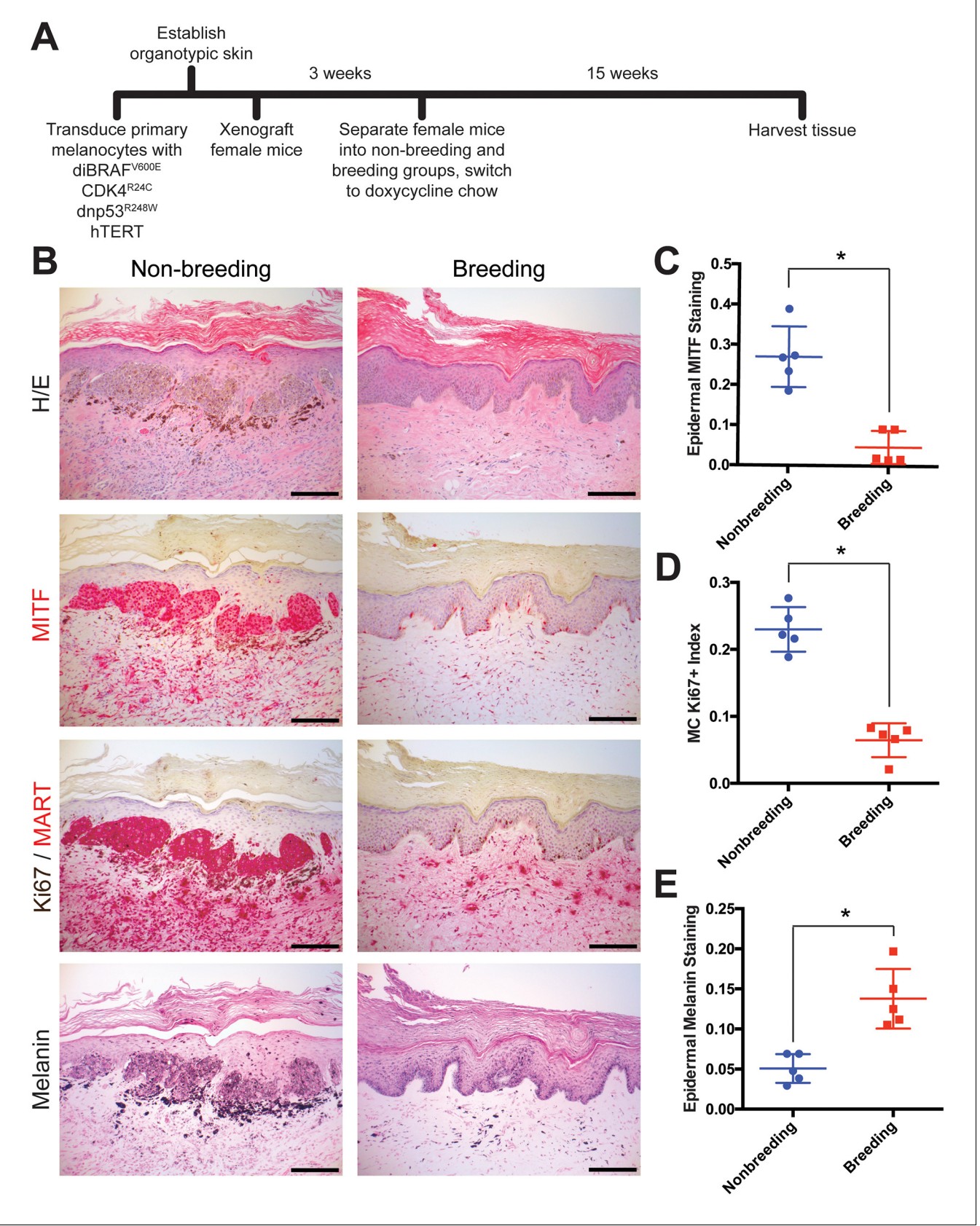

**Figure 1.** Multiple pregnancies inhibit melanomagenesis. (A) Experimental timeline of genetically-defined human xenograft melanoma on SCID mice, n = 5 per group. (B) Histologic characterization of representative orthotopic skin and resulting tumors, including hematoxylin and eosin (H/E), melanocyte and proliferation markers MITF, Ki67/MART, and Fontana Masson (Melanin). Scale bars = 100 µM. (C–E) Quantification of epidermal MITF staining (C), Ki67 proliferation index (D) and melanin staining in epidermal keratinocytes (E), * denotes significance by the Mann-Whitney test.
DOI: https://doi.org/10.7554/eLife.31770.002

The following figure supplement is available for figure 1:

**Figure supplement 1.** Multiple pregnancies inhibit melanomagenesis.
DOI: https://doi.org/10.7554/eLife.31770.003

or progesterone. Continuous estrogen exposure drove increases in melanin production, while progesterone had opposite effects (*Figure 2A*). After hormone withdrawal, progesterone treated cells quickly returned to their baseline level of melanin production. In contrast, estrogen treated cells stably produced more melanin through continual cell divisions over the subsequent 50 days. A subset of cells that were exposed to transient estrogen were subsequently treated with progesterone. This reversed the estrogen effects, and melanin production decreased to the sub-baseline level seen upon initial progesterone treatment. Remarkably, after progesterone withdrawal, these cells fully returned back to the heightened level of melanin production induced by the initial estrogen exposure (*Figure 2A*). In addition to increased melanin production, transient estrogen exposure was associated with stable increases in well-established melanocyte differentiation proteins including tyrosinase (TYR), p-CREB and MC1R (*Figure 2B*). These results indicate that estrogen signaling, even transiently, induces durable, long-lasting effects in melanocytes associated with markers of a more fully differentiated cell state.

To determine whether estrogen similarly increased melanin production and expression of differentiation proteins in melanoma cells, we treated mouse (B16F10) or several human melanoma cells (WM46, WM51, WM3702) with either estrogen, or the specific GPER agonist G-1. Consistent with changes observed in heMel cells in vivo, estrogen or G-1 decreased melanoma cell proliferation and increased melanin production, independent of the specific oncodrivers (BRAF$^{V600E}$ or NRas$^{Q61L}$) (*Figure 2—figure supplement 1A–D*). G-1 treatment resulted in a dose-dependent inhibition of melanoma proliferation, saturating at an optimal dose of 500 nM (*Figure 2—figure supplement 1E–F*). The effects of G-1 were lost completely when GPER was genetically depleted (*Figure 2—figure supplement 1H–J*). These data, coupled with the fact that G-1 is a specific agonist of GPER which has no activity on classical estrogen receptors, indicate that the entirely of the estrogen and G-1 effects in melanoma cells are mediated through GPER. Consistent with this, we did not detect expression of ER in several melanoma cell lines (*Figure 2—figure supplement 1G*). In previous work, we demonstrated that GPER was also the sole mediator of estrogen and G-1 effects in normal primary human melanocytes (*Natale et al., 2016*).

To test whether transient GPER signaling induces a persistent state in melanoma cells that affects subsequent tumor growth in vivo, we treated melanoma cells with estrogen, G-1, or vehicle in vitro, and subsequently injected equal numbers of treated cells into host mice (*Figure 2C*). Pretreatment with estrogen or G-1 markedly reduced subsequent tumor size (*Figure 2D–E*), indicating that transient GPER activation has durable, long-lasting effects on melanoma cells that limit tumor growth in vivo.

Amplification of c-Myc – a transcription factor that antagonizes differentiation and promotes proliferation, survival, and escape from immune surveillance – is one of the most common genetic alterations in human cancers, including melanoma (*Gabay et al., 2014*; *Schlagbauer-Wadl et al., 1999*). We found that GPER signaling in melanoma cells stably depleted c-Myc protein, and induced a relative growth arrest. This was associated with persistent hypophosphorylation of RB, increased expression of melanocyte differentiation proteins including TYR, MITF, and MC1R, increased expression of human leukocyte antigen (HLA), and reduced expression of programmed cell death ligand-1 (PD-L1) (*Figure 3A–D* and *Figure 2—figure supplement 1E–F*). Genetic knockdown of GPER eliminated G-1 effects on p-RB, c-Myc, and proliferation (*Figure 2—figure supplement 1H–J*). To verify this finding indicating that G-1 effects in melanoma are mediated entirely through GPER, we utilized a selective GPER antagonist, G-36 (*Dennis et al., 2011*), that specifically inhibits GPER. In melanoma cells, a two-fold molar excess of G-36 completely blocked G-1 effects (*Figure 3F*). c-Myc loss is a major mediator of the anti-proliferative effects of GPER signaling, as melanoma cells engineered to

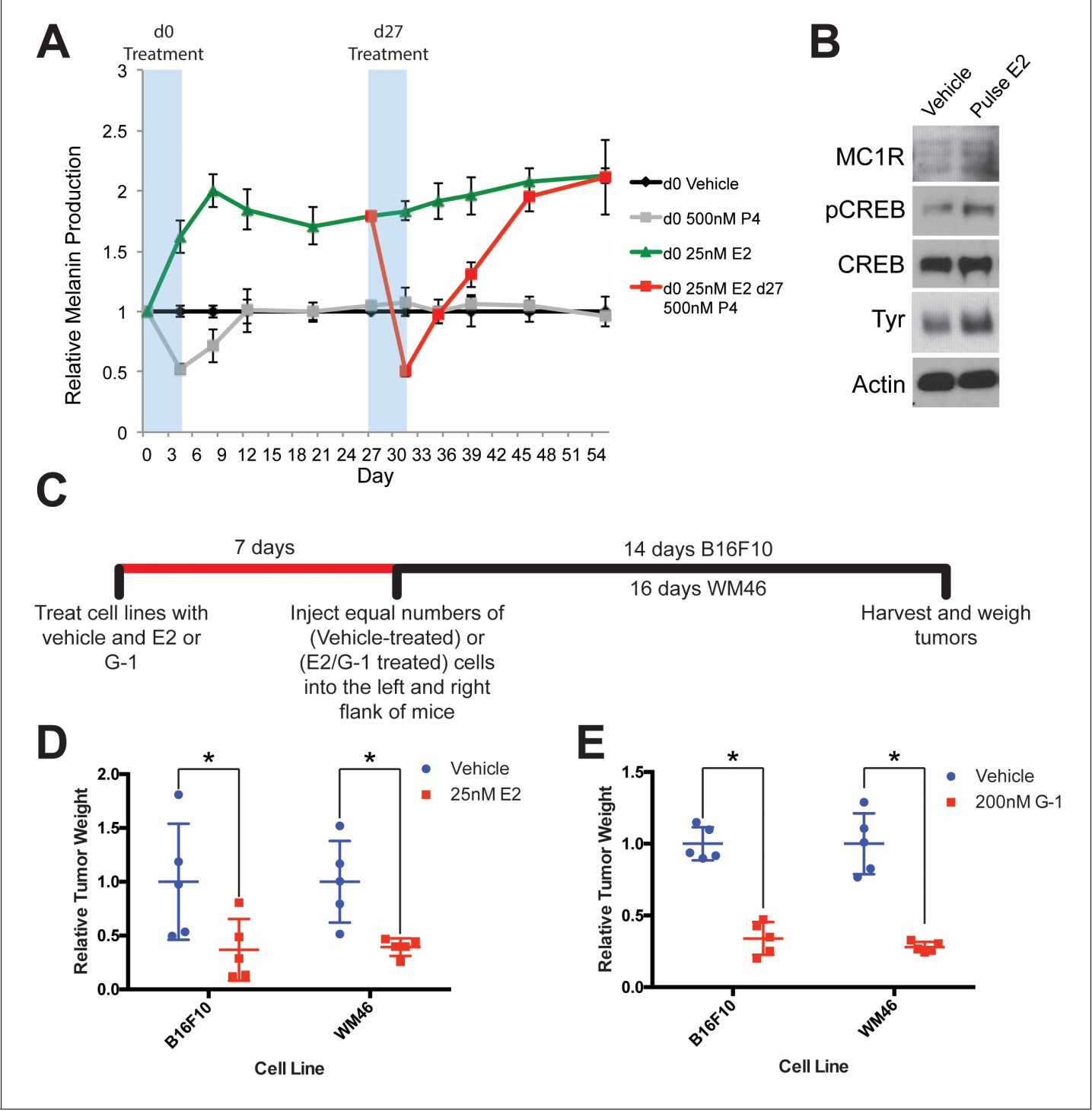

**Figure 2.** GPER signaling drives stable differentiation in normal human melanocytes and in melanoma. (**A**) Long-term melanin assay in which normal human melanocytes were transiently treated with progesterone (P4), or estrogen (E2). Subsets of these groups (Red) were treated with an additional transient pulse of P4 at Day 27. Error bars equal the standard deviation of the samples. (**B**) Western blot of melanocyte differentiation markers after a transient, 4 day treatment with either vehicle or estrogen, followed by an 8 day withdraw period. (**C**) Experimental timeline of estrogen or GPER agonist (G-1) pre-treatment of mouse and human melanoma cells, n = 5 per group. (**D**) Relative tumor weights of mouse and human melanomas pre-treated with estrogen, * denotes significance by the Mann-Whitney test. (**E**) Relative tumor weights of mouse and human melanomas pre-treated with G-1, * denotes significance by the Mann-Whitney test.

DOI: https://doi.org/10.7554/eLife.31770.004

*Figure 2 continued on next page*

*Figure 2 continued*

The following figure supplement is available for figure 2:

**Figure supplement 1.** GPER signaling slows proliferation and drives differentiation in mouse and human melanoma.
DOI: https://doi.org/10.7554/eLife.31770.005

maintain c-Myc protein in the face of GPER activation were resistant to G-1 (*Figure 3E*). c-Myc loss following GPER activation was rapid (*Figure 3G*) and PKA dependent (*Figure 3H*), suggesting that canonical stimulatory GPCR signaling destabilized c-Myc protein. Consistent with this, c-Myc loss after GPER activation was proteasome dependent (*Figure 3I*), and c-Myc protein half-life was markedly shortened (*Figure 3J*). Together, these data indicate that GPER activation regulates c-Myc through protein degradation. A recent report showed that melanomas arising during pregnancy are associated with higher GPER protein within tumor sections, suggesting that hormonal factors may upregulate GPER expression (*Fábián et al., 2017*). Consistent with these clinical data, G-1 induced a dose-dependent increase in GPER expression in melanoma cells (*Figure 3—figure supplement 1A*). To determine whether pathways downstream of GPER activated in vitro were similarly activated in vivo, we treated WM46 tumor-bearing mice with vehicle or G-1 and observed increased p-CREB and GPER, and decreased c-Myc in tumor sections (*Figure 3—figure supplement 1B–C*).

Beyond its role in stimulating proliferation and inhibiting differentiation, c-Myc was recently shown to contribute to tumor aggressiveness by promoting expression of multiple inhibitory immune checkpoint regulators on tumor cells including PD-L1 (*Casey et al., 2016*; *Kim et al., 2017*). Consistent with this, pharmacologic GPER activation in melanoma cells resulted in parallel decreases in both c-Myc and PD-L1 (*Figure 4A–C*). This PD-L1 depletion was dependent on c-Myc loss, as PD-L1 was preserved in cancer cells engineered to maintain normal c-Myc levels in the presence of GPER agonist (*Figure 3E*). Given that GPER signaling induced stable changes in tumor cells that antagonized tumor proliferation and decreased tumor cell expression of immune suppressive proteins, we next questioned whether GPER activation potentiates the anti-tumor activity of immune checkpoint blockade inhibitors which are currently the standard of care for advanced melanoma in people (*Hamid et al., 2013*; *Ribas et al., 2016*).

To determine whether tumor cell intrinsic GPER signaling influences melanoma vulnerability to immune checkpoint blockade, we took advantage of the fact that GPER effects are long-lasting. We used G-1 to activate GPER in murine B16F10 melanoma cells in vitro (*Figure 4D*). We then injected equal numbers of vehicle or G-1 treated tumor cells into syngeneic C57BL/6 mice, and treated the animals with either anti-programmed cell death 1 (αPD-1) antibody or isotype antibody control. Consistent with the premise that GPER signaling has long-lasting effects on melanoma cells, G-1 pretreatment alone inhibited subsequent tumor growth in mice and extended survival compared to controls. αPD-1 antibody monotherapy in animals injected with vehicle treated B16F10 cells also similarly prolonged survival. However, combination of G-1 pretreatment with αPD-1 antibody extended survival beyond that seen with either agent alone, indicating that GPER activity in tumor cells induced persistent changes in the tumor sufficient to improve the anti-tumor activity of systemically administered αPD-1 therapy (*Figure 4E–F*). To further demonstrate that GPER activation has tumor-cell intrinsic activity in vivo, independent of lymphocytes, we treated YUMM1.7-bearing immuno-compromised mice with G-1 (*Figure 4—figure supplement 1A*). Treatment with G-1 slowed tumor growth and extended survival (*Figure 4—figure supplement 1B–C*). Together, these data suggest that GPER signaling likely inhibits melanoma progression in a tumor cell intrinsic manner.

We next questioned whether G-1 may have therapeutic utility as a systemically delivered agent for established melanoma, with or without immune checkpoint inhibitors. Mice harboring syngeneic melanoma initiated from naïve, untreated B16F10 cells were treated with subcutaneous G-1, αPD-1 antibody, or both, and survival compared to matched mice treated with vehicle and isotype antibody controls (*Figure 5A*). G-1, which lacks systemic toxicities associated with estrogen (*Wang et al., 2009*), was well tolerated in mice, and extended survival to the same extent as αPD-1. Treatment with both αPD-1 and G-1 extended survival dramatically, indicating a marked combinatorial benefit (*Figure 5B–C*). Although B16F10 melanoma is the most commonly used model for melanoma immunology studies, and experimental results have largely translated to humans (*Benci et al., 2016*; *Twyman-Saint Victor et al., 2015*), B16F10 lacks the *Braf* or *NRas* oncodriver mutations present in most

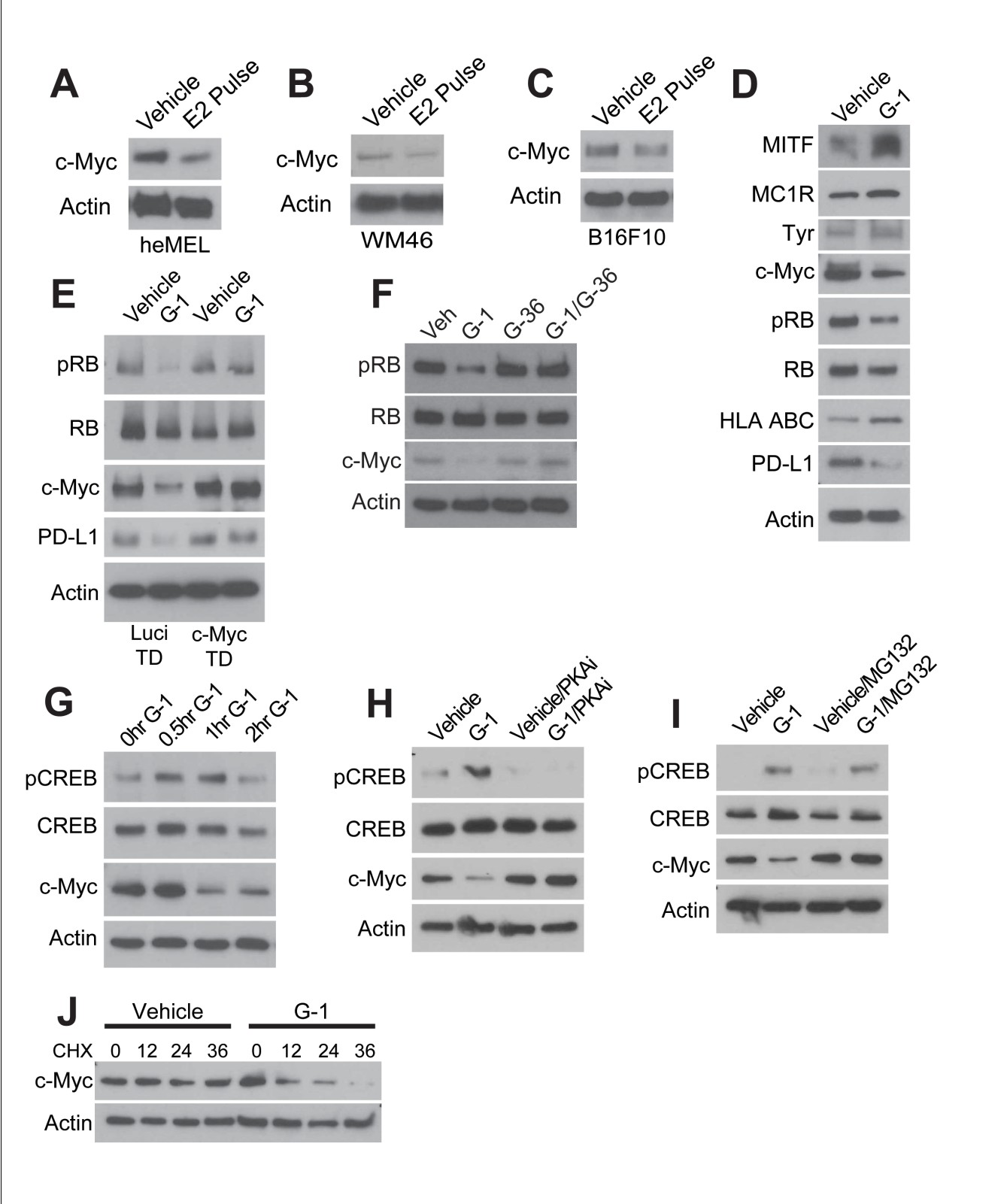

**Figure 3.** GPER signaling results in loss of c-Myc in melanoma. (A–C) Western blots of heMel (A), WM46 (B), and B16F10 (C) melanoma transiently treated with 25 nM E2 for 3 days, followed by 4 day withdraw. (D) Western blot of WM46 cells treated with 500 nM G-1 for 16 hr. (E) Western blot of luciferase- or c-Myc transduced WM46 cells treated with 500 nM G-1 for 16 hr. (F) Western blot of WM46 cells treated with 500 nM G-1, 1 μM G-36 (GPER antagonist), or a combination for 16 hr. (G) Western blot of WM46 cells treated with G-1 across a time course. (H) Western blot of WM46 cells

*Figure 3 continued on next page*

*Figure 3 continued*

treated with G-1, 100 µM PKA inhibitor Rp-8-Br-cAMPS (PKAi), or both for 1 hr. (I) Western blot of WM46 cells treated with 500 nM G-1, 2.5 µM proteasome inhibitor (MG132), or both for 1 hr. (J) Western blot of WM46 cells treated with 10 µg/ml cyclohexamide (CHX) with and without 500 nM G-1.

DOI: https://doi.org/10.7554/eLife.31770.006

The following figure supplement is available for figure 3:

**Figure supplement 1.** Markers of GPER activation in vivo.

DOI: https://doi.org/10.7554/eLife.31770.007

human melanomas (*Cancer Genome Atlas Network, 2015*; *Shain et al., 2015*). To test whether GPER signaling has similar anti-melanoma activity in a potentially more medically relevant model, we used genetically-defined melanoma cells from the newly-available Yale University Mouse Melanoma collection (YUMM). This resource contains melanoma lines generated from established genetically engineered mouse models that were backcrossed onto C57BL/6 backgrounds specifically to facilitate immunology studies (*Meeth et al., 2016*). We injected YUMM 1.7 cells (*Braf$^{V600E/wt}$ Pten$^{-/-}$ Cdkn2$^{-/-}$*) into C57BL/6 mice, and initiated G-1 treatment with and without αPD-1 after tumors reached 3–4 mm in diameter (*Figure 5D*). Similar to results observed with B16F10 melanoma, G-1 or αPD-1 monotherapy significantly extended survival, while combination treatment dramatically extended survival further, including long-term survivors (*Figure 5E–F*). These results indicate that GPER anti-tumor activity is independent of tumor oncodriver. Consistent with the hypothesis that GPER activation changes the nature of immune infiltration, G-1 treatment in melanoma-bearing mice increased several immune cell subsets within the tumors, including T cells and NK cells, suggesting a more robust inflammatory response (*Figure 5—figure supplement 1A–C*). We also observed an increase of CD8 +T cells in the central regions of tumors treated with G-1 (*Figure 5—figure supplement 1D*).

As the efficacy of immunotherapy generally decreases as the size of the tumor burden increases (*Huang et al., 2017*), we next questioned whether initiating treatment of mice with YUMM 1.7 melanoma at an earlier time point would increase the number of complete responders with long-term survival. When we began the G-1/αPD-1 regimen 4 days after introduction of YUMM 1.7 melanoma cells, the percentage of complete responders increased from 20% to 50%, with no evidence of tumor at day 100 (*Figure 6A–B*). We considered these mice 'cleared'. Cleared mice were then rechallenged with YUMM 1.7 melanoma, and we compared tumor growth and survival to age/litter matched, naïve mice injected with the same number of YUMM 1.7 cells. While the control mice grew large tumors and succumbed to disease, all of the previously cleared mice lived longer, and 80% remained tumor free without any additional treatment (*Figure 6C–F*). These results indicate that tumor clearance with G-1/αPD-1 combination therapy is associated with the formation of anti-melanoma immune memory.

## Discussion

Although five decades of clinical experience strongly suggest that female sex hormones protect against melanoma, the mechanisms through which pregnancy, or estrogen, influence melanoma have gone relatively unexplored. A pharmacologic approach that recapitulates the female/pregnancy protective effects in men, and women who have not been pregnant, might significantly diminish the overall melanoma burden. Progress in this area has likely been limited by the fact that estrogen effects in melanocytes are not mediated by the well-known nuclear estrogen receptors, but rather through the nonclassical G protein-coupled receptor GPER, which was only recently demonstrated to be expressed in melanocytes (*Natale et al., 2016*). Here, we demonstrate that this nonclassical estrogen signaling promotes differentiation in melanoma, inhibits tumor cell proliferation, and critically, promotes a phenotype that renders tumors more susceptible to immune-mediated elimination (*Figure 7*). Consistent with this, recent independent work from others has demonstrated that GPER protein levels are higher in human pregnancy-associated melanoma compared to melanoma from non-pregnant females or men, and that high GPER expression is associated with favorable prognostic indicators including decreased Breslow depth, decreased mitotic rate, and increased lymphocyte infiltration into tumor (*Fábián et al., 2017*). Conclusions from our current

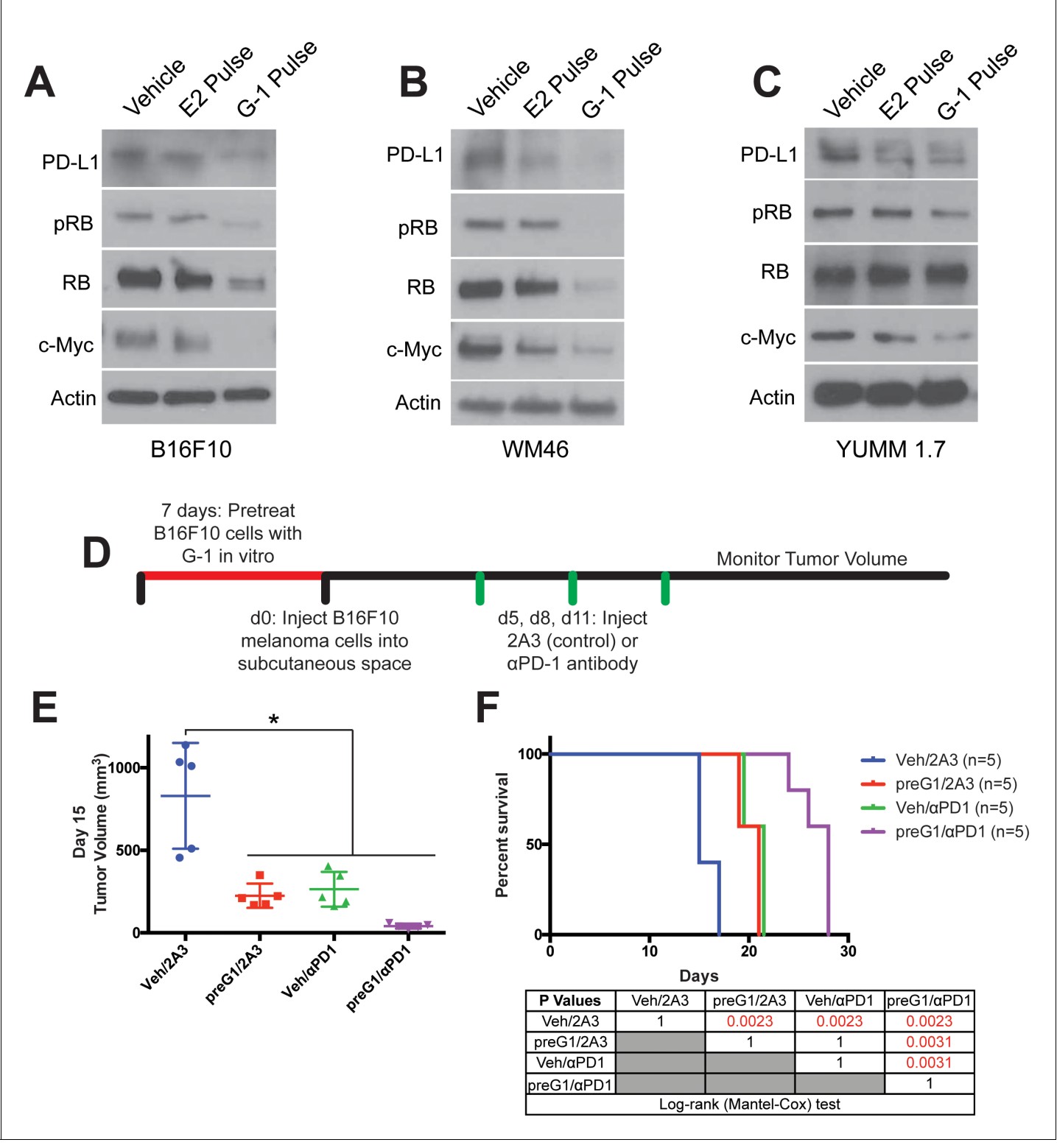

**Figure 4.** Transient GPER activation inhibits proliferation and augments response to immunotherapy. (**A–C**) Western blots of B16F10 (**A**), WM46 (**B**), and YUMM 1.7 (**C**) melanoma cells after transient treatment with a pregnancy-associated concentration of E2 (25 nM) or an optimized concentration of G-1 (500 nM). (**D**) Experimental timeline of vehicle or 500 nM G-1 pre-treatment of B16F10 cells followed by treatment with either αPD-1 antibody or isotype antibody control (2A3), n = 5 per group. (**E**) Tumor volumes of treatment groups at Day 14, * denotes significance One-way ANOVA with Tukey's multiple comparison test. (**F**) Survival curve of mice with tumors pre-treated with vehicle or G-1, followed by isotype antibody control (2A3) or αPD-1 antibody. Significance between groups by the Log-Rank (Mantel-Cox) test is listed in the table below.

*Figure 4 continued on next page*

*Figure 4 continued*

DOI: https://doi.org/10.7554/eLife.31770.008

The following figure supplement is available for figure 4:

**Figure supplement 1.** GPER activation inhibits YUMM1.7 melanoma in SCID mice.

DOI: https://doi.org/10.7554/eLife.31770.009

study are consistent with those clinical observations. While this manuscript was under review, an independent group also reported that G-1 inhibits melanoma cell proliferation in culture (*Ribeiro et al., 2017*). A second group demonstrated that GPCR signaling through the endothelin receptor influences response to targeted therapies (*Smith et al., 2017*).

We determined that one of the major mechanisms through which GPER signaling antagonizes melanoma is thorough depletion of c-Myc protein. c-Myc drives many cancers including melanoma, and despite intensive effort since its discovery nearly 40 years ago, efforts to inhibit c-Myc with systemically-tolerated agents have generally been unsuccessful, and there are still no FDA approved c-Myc inhibitors. High c-Myc protein in tumor cells inhibits expression of antigen presenting HLA/MHC (*Schlagbauer-Wadl et al., 1999*) and activates expression of PD-L1 (*Casey et al., 2016*; *Kim et al., 2017*). These combined effects of c-Myc activation render tumors less visible to immune cells. Consistent with this, GPER-induced c-Myc depletion in our study was accompanied by a reciprocal increase in HLA/MHC protein, a decrease in PD-L1 (*Figure 3D*), and an increased susceptibility to immune checkpoint inhibitor therapy.

Several lines of evidence in this work all indicate that the GPER agonist G-1 has significant tumor-cell intrinsic anti-melanoma activity. First, we show in *Figure 2* that pretreatment of mouse melanoma cells with GPER agonist in vitro drives durable cellular differentiation that inhibits subsequent tumor growth in mice. Consistent with this, G-1 pretreatment of human melanoma cells also inhibited subsequent tumor growth in SCID mice, indicating that G-1 has anti-tumor activity that is independent of CD4 +or CD8+ T cells. Further indicative of a tumor cell intrinsic effect of GPER agonist, we demonstrated that pretreatment of murine melanoma in vitro with GPER agonist still potentiated the in vivo anti-tumor activity of αPD-1 immune checkpoint blockade (*Figure 4D–F*). Finally, murine YUMM melanoma tumors established in SCID mice (lacking CD4+ and CD8+ T cells) were also inhibited by systemically delivered G-1 (*Figure 4—figure supplement 1*). Together, these data strongly support the model in which GPER agonists promote immune clearance of tumor by acting on the tumor cells themselves.

To our knowledge, this is the first work to demonstrate the potential therapeutic utility of combining GPER agonists or other differentiation-based therapy with cancer immunotherapy for any cancer type — an approach that may also prove useful for other cancers. Differentiation drivers likely have very large 'therapeutic windows' as anti-cancer agents. Melanocytes (and other GPER-expressing cells) normally respond to physiologic GPER activation, whose natural ligand is endogenous estrogen, and the synthetic specific GPER agonist G-1 is well tolerated in mice. Although no approved drugs specifically target GPER, GPCRs are biologically important and are generally highly 'drugable', as up to 40% of all FDA approved medications act through these receptors. To our knowledge, this work is the first to discover the potential therapeutic utility of combining a GPCR agonist with immunotherapy. As many tumor types express GPER, the selective agonist G-1 may ultimately prove useful in combination therapy for many human cancers.

## Materials and methods

### Cell culture and cell lines

Primary human melanocytes were extracted from fresh discarded human foreskin and surgical specimens as previously described (*McNeal et al., 2015*) with some modifications detailed as follows. After overnight incubation in Dispase, the epidermis was separated from the dermis and treated with trypsin for 10 min. Cells were pelleted and plated in selective melanocyte Medium 254 (Invitrogen, Carlsbad, CA, USA) with human melanocyte growth supplement, and 1% penicillin and streptomycin (Invitrogen). B16F10 melanoma cells were a gift from Andy Minn (University of Pennsylvania Institute, Philadelphia, PA, USA). WM46 melanoma cells were a gift from Meenhard Herlyn (Wistar

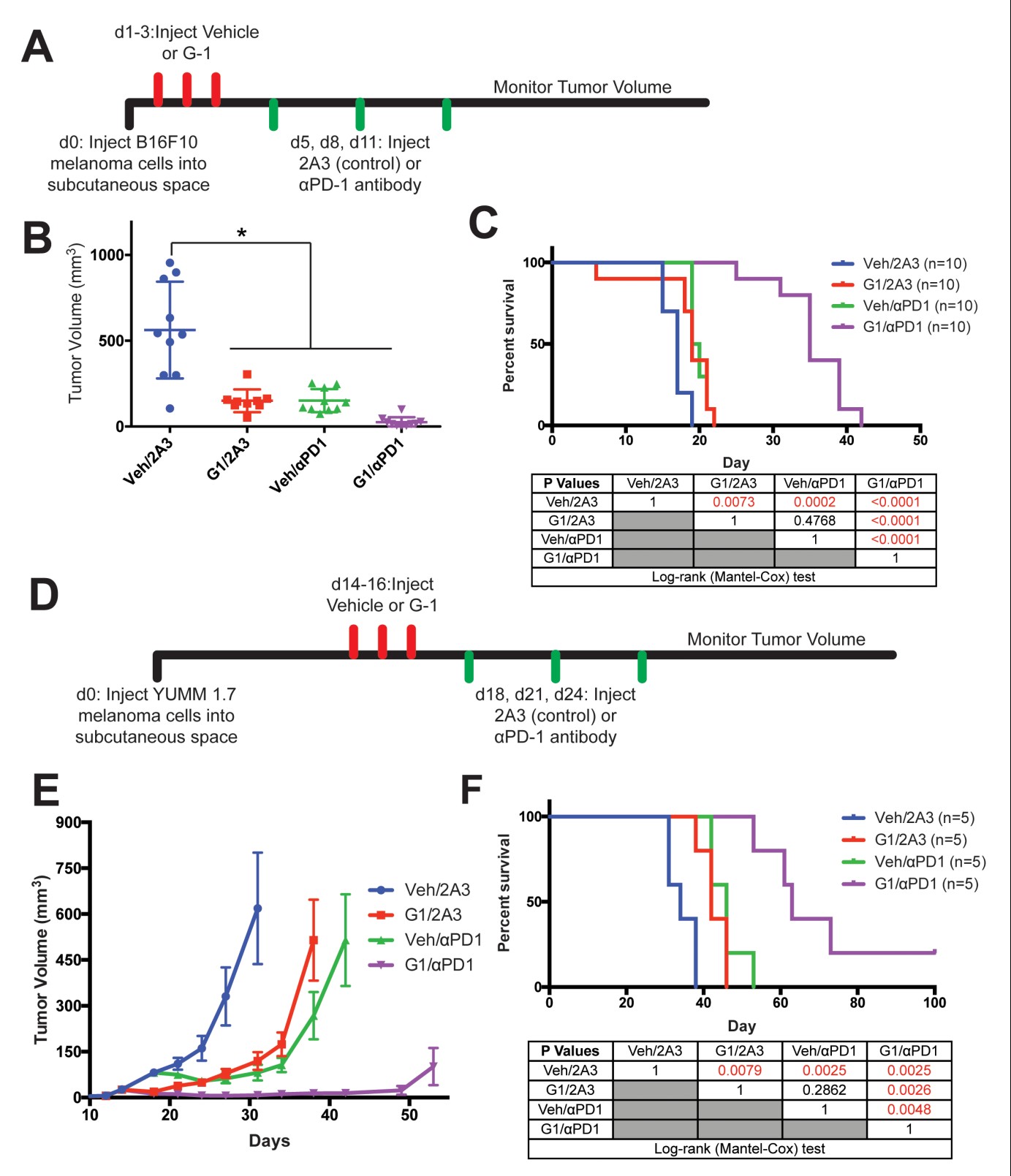

**Figure 5.** Treatment of melanoma-bearing mice with G-1 and αPD-1 immunotherapy dramatically extends survival. (**A**) Experimental timeline of B16F10-bearing mice treated with vehicle or G-1, as well as αPD-1 antibody or isotype antibody control (2A3), n = 10 per group. (**B**) Tumor volumes of treatment groups at Day 14, * denotes significance One-way ANOVA with Tukey's multiple comparison test. (**C**) Survival curve of mice treated with vehicle or G-1, as well as isotype antibody control (2A3) or αPD-1 antibody. Significance between groups by the Log-Rank (Mantel-Cox) test is listed in
*Figure 5 continued on next page*

*Figure 5 continued*

the table below. (**D**) Experimental outline of YUMM1.7-bearing mice treated with vehicle or G-1, as well as isotype antibody control (2A3) or αPD-1 antibody. Treatment was started at day 14 after tumors reached 4–5 mm in diameter. n = 5 per group. (**E**) Tumor volumes over time of treatment groups. (**F**) Survival curve of mice treated with vehicle or G-1, as well as αPD-1 antibody or isotype antibody control (2A3). Significance between groups by the Log-Rank (Mantel-Cox) test is listed in the table below.

DOI: https://doi.org/10.7554/eLife.31770.010

The following figure supplement is available for figure 5:

**Figure supplement 1.** G-1 treatment in vivo alters tumor- infiltrating immune cells.

DOI: https://doi.org/10.7554/eLife.31770.011

Institute, Philadelphia, PA, USA). YUMM1.7 melanoma cells were a gift from Ashani Weeraratna (Wistar Institute, Philadelphia, PA, USA) and Marcus Bosenberg (Yale University, New Haven, CT, USA). These cell lines were verified to be of melanocyte origin by response to alpha melanocyte stimulating hormone and melanin production. Human-engineered melanoma cells (heMel) were cultured in Medium 254, WM46 cells were cultured in TU2% media, B16F10 and YUMM1.7 cells were cultured in DMEM (Mediatech, Manassas, VA, USA) with 5% FBS (Invitrogen) and 1% antibiotic-anti-mycotic (Invitrogen). Cells were transduced with lentiviruses as described previously (*McNeal et al., 2015*). The following shRNAs were expressed from the pLKO vector and are available from The RNAi Consortium: shGPER.1 (TRCN0000026391, GAGCATCAGCAGTACGTGATT) and shGPER.2 (TRCN0000026405, GCCACGCTCAAGGCCGTCATT). Progesterone (P8783) and 17β-Estradiol (E8875) were purchased from Sigma-Aldrich (St. Louis, MO, USA). G-1 (10008933) and G-36 (14397) were purchased from Cayman Chemical (Ann Arbor, MI, USA). Rp-8-Br-cAMPS was purchased from Santa Cruz Technologies (Dallas, Texas, USA). These compounds were diluted to working stock solutions in Medium 254.

## Mice

All mice were purchased from Taconic (Hudson, NY, USA). Five- to seven-week-old female immune deficient (ICR SCID) and syngeneic (C57BL/6NTac) mice were allowed to acclimatize for one week prior to being used for experiments. These studies were preformed without inclusion/exclusion criteria or blinding, but included randomization. Based on a twofold-anticipated effect, we performed experiments with at least five biological replicates. All procedures were performed in accordance with International Animal Care and Use Committee (IACUC)-approved protocols at the University of Pennsylvania.

## Human-engineered melanoma xenografts

Organotypic skin grafts were established using modifications to previously detailed methods (*McNeal et al., 2015*). The Keratinocyte Growth Media (KGM) used for keratinocyte-only skin grafts was replaced with modified Melanocyte Xenograft Seeding Media (MXSM). MXSM is a 1:1 mixture of KGM, lacking cholera toxin, and Keratinocyte Media 50/50 (Gibco) containing 2% FBS, 1.2 mM calcium chloride, 100 nM Et-3 (endothelin 3), 10 ng/mL rhSCF (recombinant human stem cell factor), and 4.5 ng/mL r-basic FGF (recombinant basic fibroblast growth factor). Briefly, primary human melanocytes were transduced with lentivirus carrying doxycycline-inducible BRAF(V600E), dominant-negative p53(R248W), active CDK4(R24C) and hTERT. Transduced melanocytes ($1.5 \times 10^5$ cells) and keratinocytes ($5.0 \times 10^5$ cells) were suspended in 80 μL MXSM, seeded onto the dermis, and incubated at 37°C for 4 days at the air–liquid interface to establish organotypic skin. Organotypic skin tisssues were grafted onto 5–7-week-old female ICR SCID mice (Taconic) according to an IACUC–approved protocol at the University of Pennsylvania. Mice were anesthetized in an isoflurane chamber and murine skin was removed from the upper dorsal region of the mouse. Organotypic human skin was reduced to a uniform 11 mm × 11 mm square and grafted onto the back of the mouse with individual interrupted 6–0 nylon sutures. Mice were dressed with Bactroban ointment, Adaptic, Telfa pad, and Coban wrap. Dressings were removed 2 weeks after grafting and the tissue was allowed to stabilize for an additional week before mice were switched over to doxycycline chow (6 g/kg, Bio-Serv, Flemington, NJ) for 15 weeks.

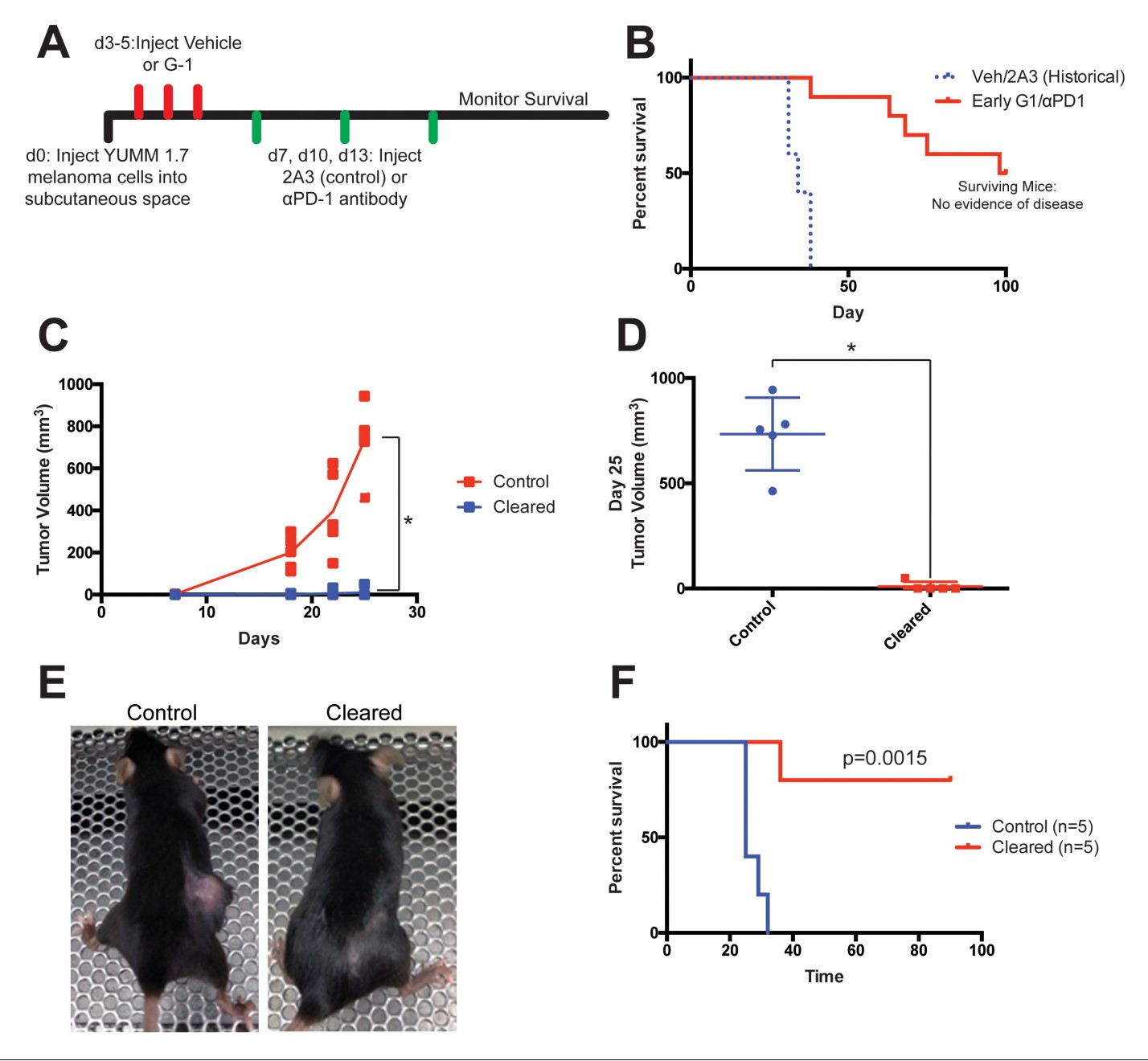

**Figure 6.** Tumor clearance with G-1 and αPD-1 combination treatment is associated with immune memory. (A) Experimental outline of YUMM1.7-bearing mice treated with vehicle or G-1 and αPD-1 antibody, treatment was started at day 3, n = 10. (B) Survival curve of mice treated G-1 and αPD-1 antibody compared to historical controls, five mice had no evidence of disease at day 100 and were considered 'cleared'. (C) Control and Cleared mice were challenged with YUMM 1.7 tumors, tumor volumes were measured over time, *denotes significance by 2way-ANOVA. (D) Tumor volumes of Control and Cleared mice on day 25, * denotes significance by the Mann-Whitney test. (E) Representative images of Control and Cleared mice on day 25. (F) Survival curve of Control and Cleared mice challenged with YUMM1.7 tumors, significance by the Log-Rank (Mantel-Cox) test.
DOI: https://doi.org/10.7554/eLife.31770.012

## Subcutaneous tumors and treatments

Subcutaneous tumors were initiated by injecting tumor cells in 50% Matrigel (Corning, Bedford, MA, USA) into the subcutaneous space on the left and right flanks of mice. For each type of tumor injection, $4 \times 10^4$ B16F10 cells were used, $1 \times 10^6$ WM46 cells were used, and $1 \times 10^5$ YUMM1.7 cells

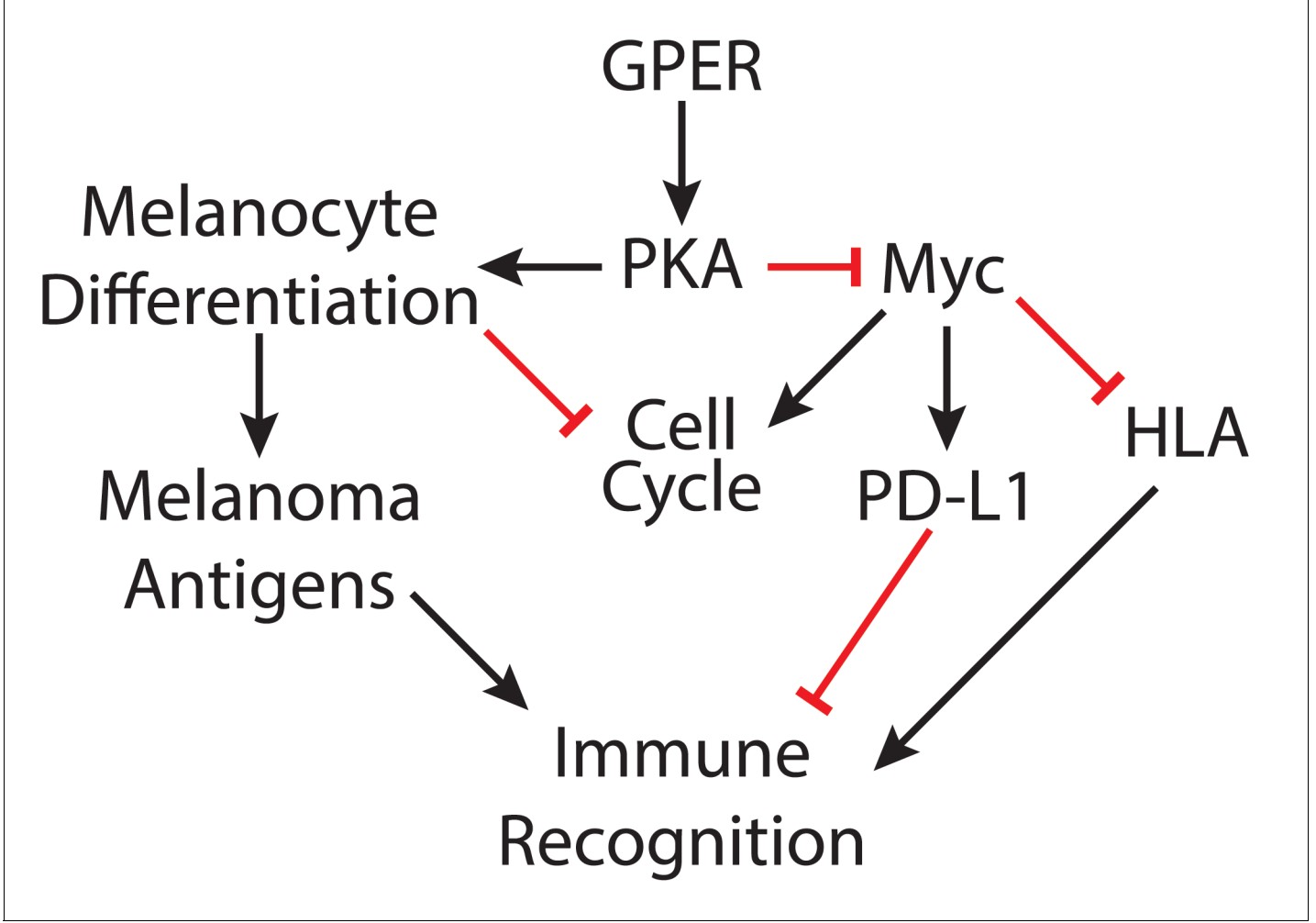

**Figure 7.** Model depicting mechanisms through which GPER signaling may antagonize melanoma.
DOI: https://doi.org/10.7554/eLife.31770.013

were used. In vivo G-1 treatments were performed by first dissolving G-1, synthesized as described previously (*Natale et al., 2016*), in 100% ethanol at a concentration of 1 mg/ml. The desired amount of G-1 was then mixed with an appropriate volume of sesame oil, and the ethanol was evaporated off using a Savant Speed Vac (Thermo Fisher Scientific, Waltham, MA, USA), leaving the desired amount of G-1 dissolved in 50 μL of sesame oil per injection at a 0.4 mg/kg dose for B16F10 experiments, and 10 mg/kg dose for YUMM1.7 experiments. Vehicle injections were prepared in an identical manner using 100% ethanol. Vehicle and G-1 injections were delivered through subcutaneous injection as indicated in each experimental timeline. Isotype control antibody (Clone: 2A3, BioXcell, West Lebanon, NH, USA) and αPD-1 antibody (Clone: RMP1-14, BioXcell) were diluted in sterile PBS and delivered through intraperitoneal injections at a dose of 10 mg/kg.

### Survival analysis

As subcutaneous tumors grew in mice, perpendicular tumor diameters were measured using calipers. Volume was calculated using the formula $L \times W^2 \times 0.52$, where L is the longest dimension and W is the perpendicular dimension. Animals were euthanized when tumors exceeded a protocol-specified size of 15 mm in the longest dimension. Secondary endpoints include severe ulceration, death, and any other condition that falls within the IACUC guidelines for Rodent Tumor and Cancer Models at the University of Pennsylvania.

## Western blot analysis

Adherent cells were washed once with DPBS, and lysed with 8M urea containing 50 mM NaCl and 50 mM Tris-HCl, pH 8.3, 10 mM dithiothreitol, 50 mM iodoacetamide. Lysates were quantified (Bradford assay), normalized, reduced, and resolved by SDS gel electrophoresis on 4–15% Tris/Glycine gels (Bio-Rad, Hercules, CA, USA). Resolved protein was transferred to PVDF membranes (Millipore, Billerica, MA, USA) using a Semi-Dry Transfer Cell (Bio-Rad), blocked in 5% BSA in TBS-T and probed with primary antibodies recognizing β-Actin (Cell Signaling Technology, #3700, 1:4000, Danvers, MA, USA), BRAF V600E (Spring Bioscience, VE1, 1:500, Pleasanton, CA, USA) c-Myc (Cell Signaling Technology, #5605, 1:1000), CDK4 (Cell Signaling Technology, #12790, 1:1000), p-CREB (Cell Signaling Technology, #9198, 1:1000), CREB (Cell Signaling Technology, #9104, 1:1000), ERα(Cell Signaling Technology, #8644, 1:1000),GPER (Sigma, HPA027052, 1:500), HLA-ABC (Biolegend, w6/32,1:500, San Diego, CA, USA), MC1R (Abcam, EPR6530, 1:1000 Cambridge, MA, USA), p53 (Cell Signaling Technology, #2527, 1:1000), human PD-L1 (Cell Signaling Technology, #13684, 1:1000), mouse PD-L1 (R and D systems, AF1019, 1:500, Minneapolis, MN, USA), p-RB (Cell Signaling Technology, #8516, 1:1000), RB (Cell Signaling Technology, #9313, 1:1000), and tyrosinase (Abcam, T311, 1:1000). After incubation with the appropriate secondary antibody, proteins were detected using either Luminata Crescendo Western HRP Substrate (Millipore) or ECL Western Blotting Analysis System (GE Healthcare, Bensalem, PA). All western blots were repeated at least three times.

## Melanin assay

Cells ($1 \times 10^5$) were seeded uniformly on 6-well tissue culture plates. Cells were treated with vehicle controls, estrogen, or G-1 for 4 days. Cells were then trypsinized, counted, and spun at 300 g for 5 min. The resulting cell pellet was solubilized in 120 μL of 1M NaOH, and boiled for 5 min. The optical density of the resulting solution was read at 450 nm using an EMax microplate reader (Molecular Devices, Sunnyvale, CA, USA). The absorbance was normalized to the number of cells in each sample, and relative amounts of melanin were set based on vehicle-treated controls. All melanin assays were repeated at least three times

## Immunohistochemistry and quantification

Formalin-fixed paraffin-embedded (FFPE) human skin tissue sections from organotypic tissue was stained for MITF (NCL-L-MITF, Leica Biosystems, Nussloch, Germany), MelanA (NCL-L-MITF, Leica Biosystems), and Ki67 (NCL-L-Ki67-MM1, Leica Biosystems). Staining was performed following the manufacturer's protocol for high temperature antigen unmasking technique for paraffin sections. For melanin staining, FFPE embedded tissue was subjected to Fontana-Masson histochemical stain as previously described (*Natale et al., 2016*).

FFPE subcutaneous tumor tissue sections were stained for CD8 (Cell Signaling Technology, #98941, 1:400), p-CREB (Cell Signaling Technology, #9198, 1:800), c-Myc (Abcam, ab32072, 1:100), GPER (Novus Biologics, NLS1183, 1:50, Littleton, CO, USA). Briefly, tissue sections were de-paraffinized, rehydrated, and subjected to heat induced antigen retrieval. Antigen retrieval was performed in 10 mM citrate buffer, pH 6.0 for CD8, p-CREB, and c-Myc; Tris-EDTA, pH 8.0 (Thermo Fisher Scientific, BP2473-1) was used for GPER. Subsequent staining procedures were performed following the manufacturer protocol from the HRP/DAB detection kit (Abcam, ab80436). Sections were counter stained with hematoxylin, dehydrated, and cover slipped with Permount Mounting Media (Thermo Fisher Scientific).

Tissue section quantification was performed according to *Billings et al. (2015)*. Briefly, 20X photomicrograph images of representative tissue sections were taken using the Zeiss Axiophot microscope and Keyence BZ-X710 (Itasca, IL, USA). Tiff files of the images were saved and transferred to Adobe Photoshop where pixels corresponding to staining were selected using the color selection and lasso selection tools. Images corresponding to the single specific color were then analyzed using FIJI (Image J) to determine the number of pixels in each sample and normalized to epidermal area. The numbers of pixels representing Fontana-Masson staining were normalized to the total amount of epidermal area. Ki67 proliferation index was calculated by dividing the number Ki67 positive cells by the total number of MelanA positive cells in the samples.

## Flow cytometry

Cell surface markers were assessed by incubating single cell suspensions of tissues with primary fluorochrome-labeled antibodies at 4°C for 60 min in PBS with 5% FBS; FITC-anti-mouse-Nkp46 (29A1.4, Biolegend, #137606, 1:50), PE-CF594-anti-mouse-CD8a (53–6.7, BD Pharmingen, #562283, 1:100), PE-Cy5-anti-mouse-CD3ε (145–2 C11, Biolegend, #100310, 1:100, PE-Cy7-anti-mouse-I-A/I-E (M5/114.15.2, Biolegend, #107630, 1:600), V450-anti-mouse-CD44 (IM7, Biolegend, #560451, 1:100), AF700-anti-mouse-CD45 (30-F11, Biolegend, #103128, 1:400), APC-Cy7-anti-mouse-F4/80 (BM8, Biolegend, #123118, 1:100), PerCP-Cy5.5-anti-mouse-CD11b (M1/70, BD Pharmingen, #550993, 1:200), BV570-anti-mouse-CD62L (MEL-14, Biolegend, #104433, 1:50), Live/Dead Fixable Aqua Dead Cell Stain Kit, for 405 nm excitation (Thermo Fisher Scientific, L-34966, 1:600). Intracellular staining was done using the Fixation/Permeabilization Kit from eBiosciences. Flow cytometric analysis was performed on LSR II Flow Cytometer (BD Biosciences). Collected data were then analyzed using the FlowJo software (Treestar, Ashland, Oregon, USA).

## Statistical analysis

All statistical analysis was performed using Graphpad Prism 8 (Graphpad Software, La Jolla, CA, USA). No statistical methods were used to predetermine sample size. Details of each statistical test used are included in the figure legends.

## Acknowledgements

The authors thank the University of Pennsylvania Skin Biology and Disease Research-based Center for primary melanocytes and keratinocytes, and for histologic processing and analysis of tissue sections, Andy Minn for B16F10 cells, Meenhard Herlyn for WM46, WM51, WM3702 cells, Ashani Weeraratna and Marcus Bosenberg for YUMM 1.7 cells, and Jeffery Winkler for synthesizing G-1. The authors also thank Sarah Millar, George Cotsarelis, Meenhard Herlyn, John Seykora, David Manning, Pantelis Rompolas and Thomas Leung for critical pre-submission review. TWR is supported by a grant from the NIH/NCI (RO1 CA163566), a Penn/Wistar Institute NIH SPORE (P50CA174523), and the Melanoma Research Foundation. CAN was supported by an NIH/NIAMS training grant (T32 AR0007465-32) and an NIH/NCI F31 NRSA Individual Fellowship (F31 CA206325). This work was supported in part by the Penn Skin Biology and Diseases Resource-based Center (P30-AR069589). The contents are solely the responsibility of the authors and do not necessarily represent the official views of the NIH.

## Additional information

### Competing interests

Christopher A Natale, Todd W Ridky: Inventor on provisional patents held by the University of Pennsylvania related to this work (PCT International Application No. PCT/US2017/035278), and cofounder of the Penn Center for Innovation supported startup Linnaeus Therapeutics Inc. The other authors declare that no competing interests exist.

### Funding

| Funder | Grant reference number | Author |
| --- | --- | --- |
| National Cancer Institute | F31 CA206325 | Christopher A Natale |
| National Institute of Arthritis and Musculoskeletal and Skin Diseases | RO1 CA163566 | Todd W Ridky |
| Melanoma Research Foundation | Established Investigator Award | Todd W Ridky |
| National Cancer Institute | P50CA174523 | Todd W Ridky |
| National Institute of Arthritis and Musculoskeletal and Skin Diseases | T32 AR0007465-32 | Christopher A Natale |

| National Institute of Arthritis and Musculoskeletal and Skin Diseases | P30-AR069589 | Tzvete Dentchev Todd W Ridky |

The funders had no role in study design, data collection and interpretation, or the decision to submit the work for publication.

## Author contributions
Christopher A Natale, Conceptualization, Data curation, Formal analysis, Supervision, Funding acquisition, Validation, Investigation, Visualization, Methodology, Writing—original draft, Project administration, Writing—review and editing; Jinyang Li, Resources, Data curation, Formal analysis, Validation, Investigation, Visualization, Writing—review and editing; Junqian Zhang, Ankit Dahal, Data curation, Formal analysis, Validation, Investigation, Writing—review and editing; Tzvete Dentchev, Resources, Data curation, Formal analysis, Validation, Investigation, Writing—review and editing; Ben Z Stanger, Conceptualization, Resources, Data curation, Supervision, Investigation, Methodology, Writing—review and editing; Todd W Ridky, Conceptualization, Resources, Data curation, Formal analysis, Supervision, Funding acquisition, Validation, Investigation, Visualization, Methodology, Writing—original draft, Project administration, Writing—review and editing

## Author ORCIDs
Christopher A Natale (iD) http://orcid.org/0000-0003-4949-3849
Todd W Ridky (iD) https://orcid.org/0000-0001-8482-1284

## Ethics
Animal experimentation: This study was performed in strict accordance with the recommendations in the Guide for the Care and Use of Laboratory Animals of the National Institutes of Health. All of the animals were handled according to approved institutional animal care and use committee (IACUC) protocol (#803381) of the University of Pennsylvania.

## Decision letter and Author response
Decision letter https://doi.org/10.7554/eLife.31770.016
Author response https://doi.org/10.7554/eLife.31770.017

# Additional files

## Supplementary files
• Transparent reporting form
DOI: https://doi.org/10.7554/eLife.31770.014

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
