## [Decision Letter]

Thank you for submitting your article "GPER signaling inhibits melanoma and improves response to PD-1 blockade" for consideration by *eLife*. Your article has been favorably evaluated by Sean Morrison (Senior Editor) and three reviewers, one of whom is a member of our Board of Reviewing Editors. The following individual involved in review of your submission has agreed to reveal his identity: Matthias Barton (Reviewer #2).

The reviewers have discussed the reviews with one another and the Reviewing Editor has drafted this decision to help you prepare a revised submission.

This manuscript describes a mechanism by which estrogen, acting via a non-canonical hetro-trimeric G-protein coupled estrogen receptor (GPER), may provide both protective and therapeutic benefits for melanoma. Using cell culture and xenograft models the authors show that pregnancy is protective against melanoma and that this effect is due to estrogen and not progesterone. The authors go on to show that this protective effect is due to GPER signaling and is dependent on PKA, indicating the engagement of an anti-melanoma GPCR->cAMP->PKA signaling axis. Activation of this GPER cascade results in decreased melanoma cell proliferation and increased expression of melanocyte differentiation markers as well as elevated expression of cell surface HLA proteins. Concurrent with the expression of melanocyte differentiation markers, tumor cells exposed to estrogen or a selective GPER agonist (G-1) showed decreased levels of PD-L1 expression. Biochemically, activation of this pathway results in decreased levels of c-MYC expression purportedly due to decreased protein half-life, dependent on PKA, suggesting the increased proliferation and loss of melanocyte markers in these cells may be driven by elevated expression of c-MYC. Finally, the authors show that combined treatment of melanoma bearing mice with a GPER agonist increases survival similarly to treatment with anti-PD-1 antibodies, and that combined treatment is well-tolerated and results in profound tumor regression, sustained responses, and in one model, development of immune memory such that re-challenge of mice with previously treated disease prevented tumor formation. Overall, this paper provides a reasonably compelling preclinical rationale for exploring GPER agonists as a potential combination with immunotherapy in melanoma.

Major concerns:

1) Results presented in Figure 3, Figure 4, Figure 5, Figure 5—figure supplement 1, and Figure 6, lack indications as to at what concentration G-1 was used in these experiments. Without this information the results cannot be conclusively assessed. In addition, related to results presented in Figure 4: What is "an optimized concentration of G-1"? What were criteria for "optimization", and how was/is "optimization" defined?

2) In Figure 3 the authors show that stabilized expression of c-MYC in WM46 melanoma cells abolishes the biochemical effects of the GPER agonist. Does the same hold true in vivo? Does this entire phenomenon really pivot on the expression of a single oncoprotein transcription factor? Furthermore, what (if any) are the effects of GPER signaling upon the melanocyte master regulator and E-box binding transcription factor MITF? This is important since MITF plays such a critical role in melanomagenesis and alterations in MITF expression can have profound effects on melanoma cell behavior.

3) There is a lack of mechanistic depth on the effects of the GPER agonists on specific immune cell subsets. Have the authors attempted to deplete either CD4^+^ cells or CD8^+^ T-cells and then repeat the animal studies with the anti-PD-1 and GPER agonist? This would add a deeper level of mechanistic depth to the manuscript.

4) As shown in Figure 2, the effects of E2 and G-1 (albeit at 8-fold higher concentrations than E2) are equally potent to suppress murine and human melanoma tumor growth in vivo. The question of interest is, if selective GPER activation is as effective as non-selective estrogen receptor activation, what (if any) relative role do ER α and ER β play in this process compared to GPER?

5) Given that the authors show increases in immune infiltration in Figure 5—figure supplement 1, a histological image showing the location of these cells in/around the tumor would be useful here. Are there CD8^+^ T-cells around the periphery of the tumors that move into the tumor upon GPER agonist treatment?

6) Note on statistical tests:

One reviewer noted concerns regarding statistical tests applied to the results of experiments. Data presented in most Figures is n>5. However, data presented in most of the panels shown in Figure 2—figure supplement 1 is limited to only n=3 observations. Yet, the authors conducted statistical analyses (two-tailed t-tests) and denote statistical significance. The authors should comment on the appropriateness of this approach since the reviewer notes that n numbers of 3 do not allow to test normality of distribution nor to select the appropriate parametric/non-parametric test. The reviewer suggests the removal of any statistical significance information from those experiments with too low n numbers. Finally, Figure 3 lacks any information as to how many replicates per experiment were performed.

[Editors' note: further revisions were requested prior to acceptance, as described below.]

Thank you for resubmitting your work entitled "Nonclassical estrogen signaling inhibits melanoma and improves response to immune checkpoint blockade" for further consideration at *eLife*. Your revised article has been favorably evaluated by Sean Morrison (Senior Editor), a Reviewing editor, and two reviewers.

The manuscript has been improved and is largely suitable for acceptance for publication but there is one small, but potentially important issue, regarding the title of the manuscript raised by reviewer #2 that needs your attention prior to acceptance, as outlined below:

*Reviewer #2:*

The authors have satisfactorily addressed the concerns raised in the previous manuscript. Some newly made changes, however, still require modification. The previous title of the manuscript was "GPER signaling inhibits […]" which has been changed to "Nonclassical estrogen signaling inhibits" in the revised version. There are several issues with this. First, any signaling can be either activating or inhibitory, thus the title either way is vague. Second, the term "Nonclassical estrogen signaling" is not used correctly here, which may be due to a misunderstanding: Classical estrogen signaling is usually only referred to as the chronic, i.e. genomic cellular events mediated by the "classical" estrogen receptors ERa and ERb. Rapid, i.e. non-genomic signaling by ERa and ERb thus also falls in the "nonclassical" signaling category. Finally, even in the absence of estrogen stimulation unliganded ERa and GPER both exert basal signaling effects due to this constitutive activity which also would qualify as "nonclassical" signaling (PMIDs 27803283 (2016) and 27888004 (2017)). For the title to be unambiguous and in accordance with the main findings presented in this paper it is suggested to change it to "Activation of GPER inhibits […]"

Further to the comments of reviewer 2, would you be willing to consider changing the title of your manuscript to:

Activation of G protein-coupled estrogen receptor signaling inhibits melanoma and improves response to immune checkpoint blockade?

---

## [Author Response]

Major concerns:1) Results presented in Figure 3, Figure 4, Figure 5, Figure 5—figure supplement 1, and Figure 6, lack indications as to at what concentration G-1 was used in these experiments. Without this information the results cannot be conclusively assessed. In addition, related to results presented in Figure 4: What is "an optimized concentration of G-1"? What were criteria for "optimization", and how was/is "optimization" defined?

Thank you for highlighting this point. We have now added concentrations of drug used to the figure legends in Figure 3, and Figure 4. Regarding Figure 5, Figure 5—figure supplement 1, and Figure 6, the preparation and administration of G-1 in vivo is also documented in the Materials and methods section under “Subcutaneous tumors and treatments”. The optimized concentration of G-1 in vitro was determined in dose response studies in Figure 2—figure supplement 1. As expected for a specific receptor mediated process, the activity of G-1 in melanocytes is saturable, and was defined as the minimum concentration of G-1 that achieved the maximum (saturable) inhibitory effect on melanoma cell proliferation. This point is now clarified in the fourth paragraph of the Results.

2) In Figure 3 the authors show that stabilized expression of c-MYC in WM46 melanoma cells abolishes the biochemical effects of the GPER agonist. Does the same hold true in vivo? Does this entire phenomenon really pivot on the expression of a single oncoprotein transcription factor? Furthermore, what (if any) are the effects of GPER signaling upon the melanocyte master regulator and E-box binding transcription factor MITF? This is important since MITF plays such a critical role in melanomagenesis and alterations in MITF expression can have profound effects on melanoma cell behavior.

We previously discussed the first point regarding c-Myc activity in vivo with the editor (email communication 10/27/2017). Melanoma cells with stabilized c-Myc are especially aggressive, and given our clear data that they do not respond to GPER agonist in vitro, they will almost certainly not respond to GPER agonists in vivo. Based on these facts, we believe (and the editor agreed) that further experiments to test the effect of stabilized c-Myc transgenes on melanoma growth of engineered tumor cells in vivo are unlikely to add significantly to the current data and are beyond the scope of this manuscript.

We agree that it would be helpful to demonstrate that key GPER induced signaling events identified in vitro also occur in vivo. To that end, we treated mice harboring human BRAF mutant melanoma tumors (WM46) with systemically administered G-1. Immunohistochemical analysis of the tumor sections showed the expected GPER induced changes in c-Myc, p-CREB, and GPER proteins that parallel changes observed in vitro. These data are presented in a new figure (Figure 3—figure supplement 1).

We agree that it is important to test whether GPER alters MITF level. We included these data in the initial submission (Figure 3) demonstrating that GPER activation induces MITF protein in melanoma cells. We have modified the text to more clearly call out this point.

3) There is a lack of mechanistic depth on the effects of the GPER agonists on specific immune cell subsets. Have the authors attempted to deplete either CD4^+^ cells or CD8^+^ T-cells and then repeat the animal studies with the anti-PD-1 and GPER agonist? This would add a deeper level of mechanistic depth to the manuscript.

We present several lines of evidence to indicate that the anti-tumor effect of GPER agonists results primarily from GPER signaling in the tumor cells, rather than host immune cells. First, we show in Figure 2 that pretreatment of mouse and human melanoma cells with GPER agonists in vitro prior to injection into host mice drives durable cellular differentiation that inhibits subsequent tumor growth in mice, without treating the mouse with G-1. Also consistent with a tumor cell intrinsic effect of GPER agonist, we demonstrate in Figure 4 that pretreatment of murine melanoma in vitro with GPER agonist still potentiates the in vivo anti-tumor activity of anti-PD-1 immune checkpoint blockade. To further address the reviewer’s question regarding the role of T-cells in the melanoma response to G-1, we introduced YUMM tumor cells into SCID mice (lacking CD4^+^ and CD8^+^ T cells) and treated animals with systemically delivered G-1. G-1 treatment in this setting inhibited tumor growth and extended survival. These data are presented in a new Figure 4—figure supplement 1. Together, these data strongly support the model in which GPER agonists act on the tumor cells to inhibit growth.

Work from others has shown that, like other cancer models, CD4^+^ and CD8^+^ T cells are required for a productive immunotherapy response to YUMM melanoma (Homet Moreno et al. 2016). We have modified the text to clarify the valuable points raised by the reviewer, and to include this reference.

4) As shown in Figure 2, the effects of E2 and G-1 (albeit at 8-fold higher concentrations than E2) are equally potent to suppress murine and human melanoma tumor growth in vivo. The question of interest is, if selective GPER activation is as effective as non-selective estrogen receptor activation, what (if any) relative role do ER α and ER β play in this process compared to GPER?

The reviewer is correct that at saturating concentrations, G-1 and E2 are equally potent at inducing differentiation in vitro prior to tumor implantation. However, the effects of both are lost completely when GPER is genetically depleted (Figure 2—figure supplement 1). These data, coupled with the finding that G-1, which is a specific agonist of GPER that has no activity on classical estrogen receptors, indicate that GPER is responsible for the entirely of the estrogen / G-1 effects in melanoma cells and primary melanocytes. As we demonstrated previously in melanocytes (Natale et al. 2016), we now added a western blot for ERa in melanoma cells, showing that the nuclear receptor is not present. It is important to note that selective GPER activation with G-1 may have significant therapeutic advantages over non-selective activation with E2, as excess, unopposed estrogen in vivo affects many tissue types with toxicities that are not observed with G-1 (Wang et al. 2009).

5) Given that the authors show increases in immune infiltration in Figure 5—figure supplement 1, a histological image showing the location of these cells in/around the tumor would be useful here. Are there CD8^+^ T-cells around the periphery of the tumors that move into the tumor upon GPER agonist treatment?

We thank the reviewers for this comment, and now include new data highlighting CD8 + T cells in treated tumor sections (Figure 5—figure supplement 1). We observed an increase in CD8^+^ T cells in the center of G-1 treated tumors.

6) Note on statistical tests:One reviewer noted concerns regarding statistical tests applied to the results of experiments. Data presented in most Figures is n>5. However, data presented in most of the panels shown in Figure 2—figure supplement 1 is limited to only n=3 observations. Yet, the authors conducted statistical analyses (two-tailed t-tests) and denote statistical significance. The authors should comment on the appropriateness of this approach since the reviewer notes that n numbers of 3 do not allow to test normality of distribution nor to select the appropriate parametric/non-parametric test. The reviewer suggests the removal of any statistical significance information from those experiments with too low n numbers. Finally, Figure 3 lacks any information as to how many replicates per experiment were performed.

Following the reviewer’s suggestion, we removed statistical significance information from panels in Figure 2—figure supplement 1 with n=3 replicates. Regarding Figure 3, in the Materials and methods section under “Western Blot Analysis”, we include information about technical replicates, indicating that each western blot was repeated at least 3 times. When possible, we have aimed to include parallel western blots in multiple cell lines to define the pathways downstream of GPER signaling.

[Editors' note: further revisions were requested prior to acceptance, as described below.]

[…] Further to the comments of reviewer 2, would you be willing to consider changing the title of your manuscript to:Activation of G protein-coupled estrogen receptor signaling inhibits melanoma and improves response to immune checkpoint blockade?

As suggested, we changed the title to “Activation of G protein-coupled estrogen receptor signaling inhibits melanoma and improves response to immune checkpoint blockade.”